# Community perceptions of citizen science approach in pandemic preparedness and response in South and Southeast Asian countries

Dinesh Kumar[1]*, Ingo Hauter[2], Felipe C. Canlas[3], Firli Yogiteten Sunaryoko[4], Gyanu Raja Maharjan[5], Md. Mazharul Anowar[6], Harjyot Khosa[7], Yi-Roe Tan[8], Peiling Yap[8]

1 Community Medicine, Dr. Rajendra Prasad Government Medical College, Sadarpur, Himachal Pradesh, India, 2 Climate Institute, Jakarta, Indonesia, 3 Wireless Access for Health Initiative, Tarlac, The Philippines, 4 Research and Development, Climate Institute, Jakarta, Indonesia, 5 Rural Development (RD) Foundation, Kathmandu, Nepal, 6 Rural Development Academy (RDA), Bogura, Bangladesh, 7 International Planned Parenthood Federation, South Asia Regional Office, New Delhi, India, 8 Health AI, Geneva, Switzerland

* dinesh9809@gmail.com

## Abstract

Citizen science (CS), a collaboration between people and scientists, is a viable approach utilizing citizens experiences in COVID-19 pandemic to manage future response. This study aimed to understand concepts, experiences, approaches, and sustainability issues of CS among citizens in five South and Southeast Asian countries. A qualitative study was carried out in Nepal, Bangladesh, India, Philippines, and Indonesia from October 2022 to March 2023. A total of 130 participants took part in four focus group discussions per country. Content analysis was done on narrative responses of participants for extraction of themes. Participants associated CS with the term "participation". Additionally, CS was related to "social responsibility" and "capacity building". During the COVID-19 pandemic, community participation was expressed by participants as a compliance measure with guidelines, helping to create awareness, and providing support (food, sanitizers, money, etc.) to fellow citizens. These experiences were related to CS and valued for personal achievement, satisfaction, and happiness with a chance to build social capital. Some participants expressed lack of confidence and language barriers as notable concerns while sharing their opinions with stakeholders (policymakers and researchers). Sustainability of CS activities was associated with creation of an organizations or networks, securing budget, incentivise participation, and assisting transportation. Participants considered CS as a community participation mechanism as a potentially viable and efficient manage future pandemics and public health crises.

**Data availability statement:** Data and code is archived at QDR (https://data.qdr.syr.edu/data-set.xhtml?persistentId=doi:10.5064/F64FIPNZ)

**Funding:** This study has been supported by Fondation Botnar (REG-20-003) through the International Digital Health & AI Research Collaborative (I-DAIR). The funders had no role in study design, data collection and analysis, decision to publish, or preparation of the manuscript.

**Competing interests:** The authors have declared that no competing interests exist.

## Background

Globally, the COVID-19 pandemic has negatively affected the economy of South and Southeast Asian countries by inhibiting socio-economic activities across regions [1,2]. Community engagement considered as a fundamental concept in the effective response and management of past epidemics [3–7]. It primarily relied on gathering participants viewpoints, experiences, shared concerns in understanding the context. It paves a way to design interventions which would be more feasible, effective, and sustainable. Citizen science (CS) comprehensively strengthens the community engagement process as a scientific approach. Vast literature is available on application of CS in domain such as ecology, environmental sciences, geology biodiversity, history/philosophy, communication, and computers/information systems [8,9]. Its role and potential benefits are explored in managing chronic diseases, vector borne diseases, rare diseases, and in healthcare settings [6,10]. CS was also used in COVID-19 with active and voluntary participation of people for efficient data collection in diverse geographical settings like Americas, Europe, and Asia [11]. Broadly, participation of citizens is viewed as an activity either to increase scientific knowledge or democratize science. Typological analysis for applications of CS has also shown a significant variation in its understanding, interpretation, and implementation. It can be considered as an activity for producing knowledge, people participation, education and learning, scientific enquiry, community-based monitoring, improving business, and assessing policy outcomes and their impacts [12,13]. For the purpose of this study, we defined CS as a practice of public participation and collaboration in all aspects of scientific research to increase knowledge, and build trust between citizens, policymakers, and researchers [14].

CS activities are largely divided into problem definition, data curation, modelling, interpretation, and communication [15]. Its role and potential benefits in managing chronic diseases, vector borne diseases, rare diseases, and in healthcare settings are increasingly being explored [6,9,10]. Currently, CS in healthcare is mostly used for data gathering processes using mobile phones and technology. For instance, it was used during the COVID-19 pandemic for quick data collection and in diverse geographical settings through active and voluntary participation of people. Its use is still limited in the co-creation of interventions, participatory modelling, data visualization, and communication [11,14]. Its limited application in healthcare, despite eagerness among citizens, could be related to the existence of a knowledge-sharing gap between citizens and researchers [15]. Given that the majority of research has been done in high-income countries, this study focused on low- and middle-income countries (LMIC). Engagement of people gives an insight through their perceptions, experiences, and possible concerns on implementation of interventions. Systematic analysis of their inputs/narratives helps in designing sustainable interventions. In our previously published mixed-method paper, the qualitative component focused mainly on potential use of digital and analogue form. It showed that actual application of digital tools substantiated the potential of digital CS in pandemic preparedness and control [16]. Current study focused on five countries where in-depth thematic analysis of

extensive illustrative quotes was done. It gave a regional context with emphasis on lived experiences of participants. Their detailed feedbacks were based on experiences during the COVID-19 pandemic to enable policymakers and researchers in the region to improve CS engagement. The specific objectives of the study were to (i) assess awareness, relatability, and knowledge to CS; (ii) assess the level of readiness in communities to participate; and (iii) identify factors (barriers and facilitators) that influence participation in related activities.

## Methods

### Ethics statement

Prior ethical approval was sought first from the United Nations University Institute, Macau (202206/01), and subsequently from ethical/regulatory committee of each country. Country specific approvals were sought from Kementerian Agama Republik Indonesia for Indonesia (B.473/In.25/LP2M/KP.01.2/07/2022), Nepal Health Research Council for Nepal (288/2022P/308), Rural Development Academy for Bangladesh (47.64.1088.014.31.206.04.448), Research Ethics Review Committee of Tarlac State University for Philippines (2022–024FCCANLAS), and Institute Ethics Committee of Dr. Rajendra Prasad Government Medical College for India (IEC/021/2022). After going through the PIS in local language, written informed consent was taken from participants by the study teams. Country-specific data are securely stored with the country team and no personal information of participants was shared while carrying out the analysis.

### Study design and site

A sequential mixed-method study was carried out in Nepal, Bangladesh, India, Philippines, and Indonesia from October 2022 to June 2023, where its qualitative segment was carried out between January 2023 to June 2023 [16]. Current study focuses on the qualitative segment of the study based on participants' experiences during the COVID-19 pandemic. It was done in Kathmandu city in Nepal, Bogura in Bangladesh, Una district of Himachal Pradesh in India, Tarlac city in Philippines, and Jakarta in Indonesia.

### Study participants

For comprehensive assessment, participants were purposively sampled across four strata for better representation of population. It included youths (18–24 years of age), marginalized and indigenous communities (people living with HIV/ AIDS, tuberculosis, malaria; ethically/socio-economic marginalized), community health workers (last mile health workers as a direct link between people and local health services), and general population (age > 24 years, non-marginalized/non-indigenous). Participants less than 18 years of age and due to inability to express their concerns, people with severe mental health conditions, or not conversant in English or the local languages were excluded.

### Selection of participants

In each country, the community-based organizations (CBOs) were purposively selected; Rural Development Foundation in Nepal, Rural Development Agency in Bangladesh, Health Applications in India, Wireless Access to Health in Philippines, and Climate Institute in Indonesia. They were already working with health care delivery system, self-help groups, and community representatives. They had a good rapport with the rural population along with indigenous and marginalized people, and community work in pandemic/outbreaks. CBOs staff selected the participants from their operational/ working areas. Based on inclusion criteria, participants were selected from respective and representative sites of their working areas. The participants were sampled from a pool of people who participated in a survey and gave consent to be re-contacted for FGD (57.2% of 2912 survey participants), as part of this mixed-method study [16]. The survey was carried out to assess participants' awareness, knowledge, and readiness to participate in pandemic related CS activities.

## Data collection

Data collection was done by the CBOs staff and were educated (graduate/post-graduate) with experience in healthcare and good working rapport with the local population. In each country, the data collection team consisted of four to six staff who identified and recruited study participants, out of which one to two staff carried out the in-person FGDs. Venues to conduct FGDs were selected by the staff to ensure convenience, privacy, and accessibility for study participants. Participants were contacted and gathered at places near their residences, such as community halls, health centres, or the office of local CBOs. As participants were familiar with these venues and the study team was from the local area, participants felt comfortable expressing their concerns freely during the discussions. In each country, four focus group discussions (FGDs) were conducted with six to eight participants per discussion for reaching to thematic saturation, i.e., no new themes are emerged from the narratives. The country teams were trained to conduct group discussions using a semi-structured guide. The guide was based on the precaution adoption process model (PAPM) and the theory of planned behaviour (TPB) [17,18]. The PAPM is used in behavioural studies where it proposes factors associated with decision-making process for an action/inaction. Whereas, TPB helps to explain effect of individual's attitude and belief on intention to act and actual behaviour. Questions were framed to gather participants' perceptions, beliefs, experiences, and concerns affecting their intention and participating behaviour in CS related activities. FGDs discussed about understanding of concept, their experiences in pandemics/outbreaks, expected nature of engagement, expected role, motivational/discouraging reasons, advantages and disadvantages, facilitators and barriers, and required resources to participate in CS related activities (S1 Text). The country teams reviewed all study materials in local languages for clarity and understandability. Participant information sheet (PIS), informed consent form, and FGD questions were translated from English to local languages (Nepali, Bangla, Hindi, Bahasa Indonesia, and Filipino) by the country teams. As the study population was more sensitive, such as the marginalized/indigenous group, the study team decided at the study design stage not to collect any demographics information. It was based on CBOs advice which was based on their past experiences while interacting with the local people. Before the conduct of FGDs, participants were shown a video and infographics in the local languages explaining the meaning of CS, levels of participation, and importance and ways of citizen engagement in various stages (pre-pandemic, alert, pandemic, and transition) pandemic preparedness and response. The levels of participation in study were; consultation by scientist, participation in planning and implementation, working closely with scientists, and empowered to define problem and implement solutions (S1 Fig).

## Data analysis

Audio recordings of the FGDs were transcribed in verbatim and translated from local languages to English. Data was analyzed manually by study teams without the use of analytical software. Microsoft Excel was used to collate and organize findings for coordination between study team members. Content analysis was done using inductive thematic analysis process by trained project staff, supervised by the site investigators. Firstly, participant narratives were selected reflecting diverse viewpoints and underlying patterns. Secondly, keywords were extracted reflecting participant's narratives. Then, codes, short words, were assigned as codes to capture core message of narratives. The initial level of coding was done by the team to observe, compare, and identify similarities and differences in the data. Subsequently, second level of pattern coding was done by the country teams. Lastly, development of themes was done by organizing codes into meaningful groups offering insight to study objectives [19]. The findings were shared with respective country teams in virtual workshops to validate analysis and coding. Triangulation of patterns was done wherein subthemes and themes were refined and grouped based on similar patterns and meaning, after obtaining consensus from teams to ensure face and content validity. It was done till saturation was obtained, i.e., no further refinement for better clarity was possible by country teams during workshops. Domains of group discussion are then summarized by the themes, subthemes, and illustrative quotes from the participants.

## Results

A total of 130 individuals participated in 20 FGDs across 5 countries (India: 27; Nepal: 25; Bangladesh: 29; Indonesia: 24; and Philippines: 25). On average, each FGD lasted for about 45 minutes (minimum: 35 minutes; maximum 90 minutes). While responding to questions, participants related to them with their perceptions for COVID-19 pandemic and subsequent response measures by their respective governments. Firstly, concepts of CS and participants' experiences and potential roles in pandemic preparedness and response were explored (Table 1):

a. Understanding of CS as a concept

Across all countries, while referring to CS, the participants collectively associated it with the term "research". Although participants were mostly (90.8%) unaware of the term but they were able to discuss their perceived concept. It was related to capacity building activity for learning and empowering themselves in the field of research. It was considered as an engagement process with an opportunity to participate in generating and analyzing data for effective implementation of pandemic control measures. It was also viewed as a social responsibility, especially in India. Distinctively, in Bangladesh, it was perceived as a way create awareness about pandemic response by the government.

b. Experiences in COVID-19 pandemic response

When participants were asked about their lived experiences during the pandemic for nature of their engagement in pandemic response, most (72.1%) of the participants (especially in India and Philippines) stated their participation by adhering to government guidelines in terms of following COVID-19 appropriate behaviour like wearing masks, maintaining physical distancing, and hand sanitization. They complied with digital interventions especially downloading mobile-based applications to give personal information to assist the authorities in tracing the spread of infection. Largely (57.7%) in India, participants said that they proactively came forward to help fellow citizens to support by providing food, medicine, masks, and sanitizers. They also worked as a part of a team with the local health authorities in organizing community-based awareness activities to provide knowledge about disease, its control measures, and importance of vaccination to reduce disease transmission.

c. Potential roles in pandemic preparedness and response

After reflecting on their experiences, participants were asked about their views on possible role(s) in future pandemics. Across all countries, based on their lived experiences, most (82.7%) considered their likely participation for assisting/providing food, masks, and sanitizers along with creating awareness about pandemic in communities. Participants also stated that as citizens, their role can be to ensure compliance by fellow citizens to government guidelines. Some (20.0%) stated that they can even contribute, based on their skill sets (e.g., computer skills for data entry), by doing some work to assist pandemic preparedness and response.

After showing participants the concept and meaning of CS using infographics, participants mentioned advantages and disadvantages of CS and highlighted potential facilitators and barriers to their participation in CS (Table 2):

a. Advantages and disadvantages of CS

While speaking about advantages, across all countries, participants expressed that CS is a people-centric approach that can be used to understand reality on the ground, gather people to encourage exchange opinions, and identify people-led solutions. Potentially, CS can build the capacity of people by generating knowledge and creating awareness about their health and pandemic control measures. The disadvantages of CS were largely linked to individual level factors such as lack of capacity or unwillingness to share opinions that will likely cause ineffective participation. Based on their COVID-19 experience, participants expressed a potential risk of infection to themselves and their families while gathering people in CS activities.

**Table 1. Thematic analysis of citizen science (CS) concept and participants' potential roles in CS along with lived experiences during COVID-19 pandemic in South and Southeast Asian countries, 2022-23.**

| Theme | Subtheme | Illustrative quotes by the participants |
|---|---|---|
| **Understanding of CS as a concept** | | |
| Capacity building in research | Learning valid methods | "*Based on the assessment, it is necessary for citizens to learn the proper methods and actions for the betterment of the community.*" (Community health worker, Philippines) |
| | Feeling empowered | "*Citizen science involves a society that wants to take an active role in conducting research with the aim of empowering themselves and others around them.*" (Youth, Indonesia)<br>"*[…] They are working as scientists in taking care of the community. For me, citizen science has given the opportunity to people to do research in a scientific way even if they don't have any scientific background.*" (Community health worker, India) |
| Participation in research | Data generation and analysis | "*Science itself has limitations whether it is geographic, ability, or data limitations but involving the community in a large number automatically gives the data. Having plenty of observations in research is also better at seeing whether there is a pattern or not, or you can conclude a phenomenon with a large data set more precisely.*" (Youth, Bangladesh)<br>"*For me the word that stands out is 'public research and participation' in which people can contribute to the collection, interpretation, and data analysis […].*" (Community health worker, India) |
| | Active engagement | "*Citizen science is the key to an active involvement of the community, especially in engaging directly in the research process, and developing science collectively.*" (Marginalized, Indonesia)<br>"*I would say it is an attempt to get the results of a research which aims to solve a problem or get a solution by directly involving people in research. Rather than just on theory, citizen science is testing directly on the community, where the community is reinvented in a research design.*" (Youth, Bangladesh) |
| Social work | Social responsibility towards each other | "*The 'social' word comes to my mind from citizen. Citizens are from our society only and according to me, citizen science itself means social work done by people together. So, it will be social it will be for society. All the human beings who are around us are working together.*" (General population, India) |
| Knowledge | Creating awareness | "*I think citizen science means publicity and awareness.*" (Community health worker, Bangladesh) |
| **Experiences in COVID-19 pandemic response** | | |
| Compliance | Adherence to government's advisories/ guidelines | "*You are asking about involvement, so we got involved in the activity by strictly following the guidelines. When they said stay at home, we exactly stayed at home, we followed them, so this is also an involvement of ours. We asked to make a distance, made a distance, asked to wear a mask, wear a mask. So, if we have done all this then we have made our contribution.*" (General population, India)<br>"*Pandemic was an accidental thing. Nobody knew when it came, and now we had an experience of a pandemic and we learned what to do? In such conditions, we should stay safe in our homes, not go outside without any reason, not go to meet others, and use a mask when we go outside (it also prevents contact with dust and smoke, and reduce transmission due to sneezing and coughing).*" (Marginalized, Nepal) |
| | Sharing information using mobile applications | "*By downloading the Aarogya Setu application. We have shared information with the government by answering the question asked in the application.*" (Youth, India)<br>"*I myself want to add an activity that maybe some of you are quite familiar with because I assume most of you have it on your smartphone. The application's name is Peduli Lindungi and it helps Indonesian citizens to track Covid development (spread) from the provincial to local level.*" (Marginalized, Indonesia) |
| Support and assistance | Distribution of essentials (masks, sanitizer, food) | "*In this (COVID-19 pandemic), people made and distributed masks, prepared and distributed food, and many volunteers came forward, who sanitized their societies (residential), or gave every facility to the people like food, etc. when someone's reports came positive, he/she himself/herself or his/her family member cannot go outside for purchasing food/ration. Then, many people came forward to help them by supplying food/ration to their houses and thus prevent the spread of the infection.*" (Marginalized, India)<br>"*Yes, there were food donations, and we cooked and distributed them to each household […].*" (Community health worker, Philippines)<br>"*During COVID-19 I was involved with the local community/ neighbours by providing rations and money to the affected families.*" (Community health worker, India) |
| | Medicine delivery for chronic diseases | "*[…] Also, for the seniors who couldn't go out for their vaccination and maintenance, we collected their cards and took them to the health centre or RHU (Rural Health Unit) to get their medicines. We delivered the papers to each house.*" (Community health worker, Philippines) |

*(Continued)*

Global Public Health

**Table 1.** (Continued)

| Theme | Subtheme | Illustrative quotes by the participants |
|---|---|---|
| Creating awareness | Disease spread and control | "*During pandemic many people have done work on how to stop it. Some educate the community that how we can stop it in a good way, like wearing a mask, keeping distance from others.*" (Marginalized, India) |
| | Importance of vaccination | "*For me, madam, it's about vaccination. We taught people about the importance of getting vaccinated because it provides protection against whatever it is. They need to get vaccinated for their safety, at least to some extent.*" (Community health worker, Philippines) |
| **Potential roles in pandemic preparedness and response** | | |
| Support and assistance | Distribution of essentials (masks, sanitizer, food) | "*Keep masks and sanitizer to their (needy) homes, they know about the risks but they were careless (not active) so that they did such things (not purchased).*" (Marginalized, Nepal) |
| | Data entry | "*If I want to do social work for the long term then I would like to do data entry. If any helping hand is there who can bring me the data, then it will be easy for me to do data entry.*" (General population, India) |
| Creating awareness | Disease spread and control | "*I will make the community understand about the diseases because most of the diseases transferred in lack of information.*" (Marginalized, Nepal) |
| Compliance | Adherence to government's advisories/guidelines | "*To follow the guideline and health protocol, maintaining discipline is a key trait in controlling the flow of the pandemic.*" (Community health worker, Indonesia) |

b. Facilitators and barriers to participation in CS activities

Participants expressed individual-level features as a potential facilitating factor for their participation. In India, CS activities are expected to instil a sense of achievement, satisfaction, and happiness. In Philippines, potential chances to gain new knowledge and build social networks were considered to be facilitators. Whereas, in India and Nepal, majority (69.2%) of the participants expressed their participation as a sense of social duty and service during a pandemic.

Similar to facilitators, individual-related factors were also considered as barriers to CS activities. Nature of concerns varied across countries, such as the lack of information in Bangladesh, Indonesia, and Philippines; lack of education, confidence, and individual level conflicts in India and Nepal; while lack of time to participate was expressed as a barrier across all countries. Participants also mentioned that the nature of facilitation can have an impact on the level of participation. In Nepal, participants said that they tend to feel demotivated if other participants choose not to participate, while one is making an effort. Also, factors like unengaged organizers/facilitators and lack of transportation assistance can pose barriers to participate.

Finally, factors related to interactions with stakeholders (Table 3) and sustainability and resources needed (Table 4) to foster strong public partnerships for pandemic preparedness and response were discussed:

a. Factors related to interaction with stakeholders

When asked about factors to consider when interacting with stakeholders such as policymakers and researchers, most (67.1%) factors were related to individual characteristics, effective communication, and provision of feedback. Active participation and a strong sense of collectiveness were raised as important factors. In India and Indonesia, contextual awareness of stakeholders and their ability to correctly understand participants were shared as likely factors. Participants expressed their lack of confidence while sharing opinions with stakeholders due to differences in their level of education and language which can result misunderstandings. In Indonesia, participants expressed concerns about use of their feedbacks, as they shared their opinions, data, and information in various activities like surveys but they do not know about their utilization.

**Table 2. Thematic analysis of perceived issues (advantages, disadvantages, facilitators, and barriers) of citizen science (CS) to foster public partnerships for pandemic preparedness and response in South and Southeast Asian countries, 2022-23.**

| Theme | Subtheme | Illustrative quotes by the participants |
|---|---|---|
| **Advantages of CS** | | |
| People-centric | Realistic approach | *"They can gather more data or the pulse of the people. Because if you are just on top, you won't see what's happening below. But if you involve the people, you will know the actual situation in the community. You can get a better feel for the pulse of the people."* (Marginalized, Philippines) <br> *"Particularly people are familiarized with challenges and aspects that are affecting their daily lives and matters relating to their own intentions. How activities can be tiered to data and processed to make a policy."* (Youth, Indonesia) |
| | Sharing of views | *"Like-minded people group will be formed, people will come forward and everyone's views will be listened to."* (Youth, India) |
| | Identifying solutions | *"Because the answers of the public can lead to a solution. So it's really necessary if we are somehow connected."* (Youth, Bangladesh) |
| Capacity Building | Gaining knowledge | *"The advantage of involving them is that they (people) gain knowledge about health and the right steps to take for their well-being. They learn what is best for them."* (Community health worker, Philippines) <br> *"It is necessary to involve the public in these activities to inform because most of the people didn't know the information about the pandemic."* (Marginalized, Nepal) |
| | Creating awareness | *"Community awareness is the advantage of involving the public in research."* (General population, Philippines) |
| **Disadvantages of CS** | | |
| Ineffective participation | Not sharing of opinions | *"People keep the solution of a problem with them only and not share it with anyone."* (Youth, India) |
| | Lack of capacity | *"My concern is that science works a bit differently so far because only people that are skilful are doing it. And not all people will understand the results of the research, people don't know about it at all."* (General population, Indonesia) |
| Risk of infection | Non-adherence to guidelines | *"Since the virus has unknown health implications for different groups, it is advisable to not involve the public actively in the early stage."* (Community health worker, Indonesia) <br> *"Social distancing, because we cannot socialize, it is difficult to socialize with other people because there is a risk of getting infected, things like that."* (Youth, Philippines) |
| **Facilitators for participation in CS activities** | | |
| Individual factors | Achievement | *"One of the factors is a sense of achievement that my voice is being listened to and implemented."* (Youth, India) |
| | Satisfaction | *"Emotional, feeling empowered in control of an issue when you join here because you feel like 'I joined here' like I was able to help the community, it is like you feed your ego and now you are satisfied."* (Youth, Philippines) |
| | Happiness | *"As we are still young, at this age, I would like to help others, because we get eternal happiness in helping others. If someone praises us or if someone gets cured due to my help, then I will be very happy."* (General population, India) |
| | Gain new knowledge | *"For me, the first main reason is that hopefully, this research can bring a change for us in the future. The second reason is that this is a recipe ingredient for researchers in terms of operational development which will be required later, and the last thing is that we can get knowledge too. In citizen science, usually, the results we can read for our reference and increase our knowledge too."* (General population, Indonesia) <br> *"To be able to learn new knowledge and skills. To acquire new learning."* (Marginalized, Philippines) <br> *"I see the benefits, the second is to add insight and knowledge […]."* (Community health worker, Indonesia) |
| | Build social networks | *"[…] And the third is that we carry out social activities to increase relations and networking."* (Community health worker, Indonesia) |
| | Social duty and service | *"It is a social duty. I am an emotional person, so I use to participate in such kinds of activities. Popularity is minor for me."* (Youth, Nepal) <br> *"Main thing is social service. Human is social animal so he/she searches for social service. Other thing is, cultural, and attitudes are also related to social."* (General population, Nepal) |

*(Continued)*

**Table 2.** (Continued)

| Theme | Subtheme | Illustrative quotes by the participants |
|---|---|---|
| Barriers to participation in CS activities | | |
| Individual factors | Lack of education | "*Lack of education which creates trouble to make them understand.*" (Marginalized, Nepal) |
| | Lack of Information | "*The main reason for not participating may be because of the lack of information.*" (Youth, Indonesia) |
| | Lack of confidence | "*Lack of initiative. A person itself does not want to help anyone or put his/her personal opinion in front of anyone.*" (Youth, India) |
| | Lack of time | "*As we are working women, we have to come to the office from 10 AM to 5 PM, before that we have to do household work, look after the children, then when we reach home in the evening, we have to prepare food, taught children, etc. So, we do not get time to participate in such activities, it pushes us back.*" (Marginalized, India) <br> "*Yes, if I am, maybe the most important reason is that the first is a matter of time. For people who work (job), there are time-related problems, scheduling problems, and how long for example we will have to participate. Don't mind doing it during free time outside working hours.*" (General population, Indonesia) |
| | Demotivation | "*If I get motivated by something then also get discouraged by that thing, like if I do some social work, many people are helping in that and many do not. Those who do not help, then I feel discouraged that if they are not helping, then why should I do the same?*" (General population, India) |
| | Risk of infection | "*I wanted to help during Covid, but I had a fear that I might get infected. If I had it, then the small children in my house may also get infected. So, I did not participate because I was afraid that this thing might happen to me and my family.*" (Marginalized, India) |
| Organizational factors | Unengaging organizers | "*Because there are organizers who are boring when teaching. And sometimes I do not know if they are just lazy to teach or if they don't want to teach what they know. The same goes for me, I prefer someone who is lively because being a killjoy is not allowed here.*" (Marginalized, Philippines) |
| | Lack of transportation | "*For me, an example would be when the travel distance is far, and there is no immediate transportation available to get off, but there must be a way, but that is the most common reason for me, the distance, the location.*" (Community health worker, Philippines) |

**Table 3. Thematic analysis of factors related to interaction with stakeholders for citizen science (CS) to foster public partnerships for pandemic preparedness and response in South and Southeast Asian countries, 2022-23.**

| Theme | Subtheme | Illustrative quotes by the participants |
|---|---|---|
| Factors related to interaction with stakeholders | | |
| Individual factors | Active participation | "*We have to work together along with policymakers, researchers, political leaders, and other stakeholders with active participation.*" (Marginalized, Bangladesh) <br> "*What I want to emphasize here is that Indonesia must have a strong collectiveness for us to interact with researchers or policymakers.*" (Marginalized, Indonesia) |
| | Lack of confidence | "*I am worried that when I interact later with researchers, my intention is to interact with them instead of being misunderstood and creating anything that actually creates unnecessary misunderstandings later.*" (Youth, Indonesia) |
| Effective Communication | Understanding of local language | "*First of all, not everyone knows Hindi very well and if we speak in our native language, then we think whether they will understand or not and if there is an educated person there, then there is a little fear.*" (Marginalized, India) |
| Provision of feedback | Utility of participation | "*When it comes to worries, sometimes there are worries that arise because of small things like for example we come to a government office or we help researchers, keep asking for feedback and suggestions or surveys like that. There are also some surveys sometimes that are done for the sake of formality, the formality of the activity, but they would not contribute to any improvements in the future. So, the worry is that we are tired of giving input, giving suggestions, giving opinions, but it is not being maximized properly.*" (General population, Indonesia) |

**Table 4. Thematic analysis of perceived sustainability factors of and needed resources for citizen science (CS) to foster public partnerships for pandemic preparedness and response in South and Southeast Asian countries, 2022-23.**

| Theme | Subtheme | Illustrative quotes by the participants |
| --- | --- | --- |
| **Factors for sustainability** | | |
| Organizational | Mechanisms of engagement | "*If I am talking about the future of citizen science, will it be sustainable or not? I think it all depends on what? How? In Indonesia, the cultural factor is very strong. And this is something that is different in each region. For example, in areas with a strong mutual cooperation system, there is a possibility that citizen science can be sustainable in the long term. However, in an area where people tend to be individualistic, this citizen science may be something that is difficult to exist in the long run. So, whether yes or no is relative, even if for example we all depend on this model, how do we involve the process from the bottom up. But usually for large-scale research like this, for example, we use a model that is almost the same in every country. If we want to make a specific model for each region, it might seem even more difficult, maybe it will take a long time.*" (Youth, Indonesia) |
| | Support | "*If we want to do such work, we cannot do anything alone. Some resources may be required like incentives, training, increased attendance and services of community health workers in the health sectors or community clinic including vehicles.*" (Community health worker, Bangladesh) |
| | Incentives | "*If we get an incentive for these activities then we can do it properly because in today's time if we want to do any work then we need money for that.*" (Youth, India) |
| | Availability of transportation | "*Being a woman, I would prefer to work in working hours which should be flexible. Pick and drop facility should be there.*" (Youth, India) |
| Culture | Values | "*Motivational factors in terms of society, ethical process, not only money. So, we need to conserve our culture to make aware society. Citizen science needs to embrace humanity then it will be sustainable.*" (Community health worker, Nepal) |
| Formalization | Subject for education | "*I think that we should promote this thing at a state level and in the education system as a vocational subject, according to a current literacy level, we will be able to collect good data in it, in every way.*" (Community health worker, India) |
| Research | Evaluation and refinement | "*For sustainability, as we have it in the plan such as PDCA (Plan-Do-Check-Act). So, if there is an improvement, then we bring it into action, to bring it into action, we put a person in charge so that he actively and further what we should do, we can send a requirement for it if it does not happen, then we will re-plan it, how to be better next time.*" (Community health worker, India) |
| **Resources needed for CS** | | |
| Organizational | Separate body | "*An organization should be there, where people who want to work in such activities can get registered themselves.*" (General population, India) |
| | Infrastructure | "*Separate centres should be opened and separate training should be given for that and trainers should be appointed.*" (Community health worker, India) |
| Finance/Leadership | Provision of funds | "*I think stable financing from private sectors as well as grants from other countries is a key in sustaining the system over time. Needs also to be backed-up by strong leadership, especially in Indonesia a strong leadership figure is required to force change for a better world.*" (Community health worker, Indonesia) |

## b. Sustainability factors and resources needed for CS

When asked to express potential sustainability factors, the importance of local mechanisms to sustain people's participation in CS activities was mentioned in Indonesia. The need for support of health system, provision of incentives (monetary and non-monetary), and transportation assistance, especially to women, were delineated in India and Bangladesh. Contextually, in India and Nepal, cultural factors were expected to play a vital role in the sustainability. In India, introduction of CS in formal education system as a distinct identity was also suggested as a sustainability factor to teach its concepts and applications. A participant from India suggested that the use of a scientific approach to monitor and evaluate CS activities for concurrent refinements can further enhance its sustainability. A formal structure such as the establishment of an organization or network and creation of infrastructures (centres) along with budgetary support were considered as resources needed for sustainability.

## Discussion

Current study showed that participants were not aware of the term citizen science but were aware and relate themselves with participatory activities. Based on their pandemic experience they expressed their readiness to participate in CS related activities. They expressed social culture and individual's motivation and attitude as facilitating while lack of incentive/support as a limiting factor. Current study captured COVID-19 lived experiences and qualitatively explored the potential application of CS approach to encourage people's participation in pandemic preparedness and response. We attempted to address contextual sensitiveness associated with LMICs, particularly South and Southeast Asian countries as well as population representativeness by including diverse groups such as youth, marginalized and indigenous groups along with the general population and community health workers. This adds value to the current literature which mainly focus on participants who have the capacity and accessibility to participate in such studies, thus leaving out hard-to-reach populations [19]. In our previous study, qualitative quotes were also used to triangulate the main quantitative findings on perceived factors influencing participation using digital as compared to analogue method. Moreover, it was done in nine countries with largely quantitative analysis focusing awareness, readiness and feasibility of citizen science. It primarily assessed the stage of readiness of participants in each country. It gave a general overview for policymakers and researchers that what to focus if they want to move people to readiness levels [16]. In current paper, analysis is focused on five countries of region where an in-depth thematic analysis on extensive illustrative quotes based on their lived experiences was done. It gave practical recommendations with contextualized insights which are actionable for policymakers and researchers in the region to improve citizen science engagement for future pandemics. The lived experiences presented in this study explored the barriers and facilitators through the lens of the participants' COVID-19 experiences, as well as participants' concerns while interacting with stakeholders (researchers and policy makers).

Thematic analysis showed that CS was not a commonly known term among the participants, but after being shown the infographics, it was expressed as a social concept, especially in India and Nepal. Across all countries, participant expressed that participation was related to individual level factors. Findings from this study resonated with the literature, where issues like self-esteem and self-efficacy along with health status were observed to be related with participation [20,21]. Current study expressed that individual viewpoint towards activity, considering a social responsibility, risk for being infected, and nature of pandemic also influence their readiness to participate. Subjective norms, cognitive ability of participants, experience of disaster, and perception of risk were delineated as individual-level factors for participation [22]. Current study showed that lack of awareness/education, time, and transportation, and distant geographical locations limit their participation. Evidence has shown that level of participation is affected by availability of service, geographic location, and socio-economic status of community [23]. Current study showed that people adhered to government guidelines like following COVID-19 appropriate behaviour and sharing personal data on mobile phone applications. Although, issues related to data security and internet availability were raised as concerns. A study showed that community engagement in emergencies was observed to be improved with accessibility to internet. It showed an improvement in sustenance of availability of health care services [23]. Current study showed participation in CS activities can be sustained by establishing formal organization support, management application, and securing funds. Evidence also showed that enhanced community participation is affected by an established management system, capacity building, and experiences and vulnerability of disaster [24]. Current study summates that COVID-19 experiences gave rationale and ways to raise resources to conduct CS activities for meaningful and effective participation.

CS makes public participation in science more democratized while involving researchers makes it more rational and objective [25]. Current study showed that people being the recipients of policies and programs, ready to have their voices included. It also showed that to provide local resonance with the CS approach, it is vital to consider the contextual settings while studying the values, norms, and culture. Individual capacity in terms of baseline knowledge and skill, awareness, and cognition needs to be mapped to tailor engagement efforts. It is also evident from current analysis that people are ready to participate but they need to be informed and their fears (infection, miscommunication, and being misunderstood

by stakeholders) needs to be addressed. Efforts need to be targeted to make people comfortable and respected without any language barriers. Measures need to be in place to identify/foresee potential conflicts between people and train for conflict resolution strategies to improve participation. A formal feedback structure should also be available to inform the public about the utility of their participation and the inclusion of their opinions in formulating interventions and policies. It is evident that feedback increases the level of motivation and it should be specific with reasons or criterion-based [26].

Current study has its strengths in terms of in-depth examination with a structured inquiry. It had an optimal sample size with rich narratives/quotes from representative population (youth, marginalized and indigenous, and general) along with community health workers. It covered both provider and beneficiary perspectives with their lived experiences during the COVID-19 pandemic. It provided relatable responses towards pandemic preparedness and control. Similar and dissimilar views of participants were analyzed across five countries. Although generalization of qualitative information poses one of the limitations along with management of voluminous data [27]. Another limitation is lack of participants' demographic information to support contextual findings. Nonetheless, it has helped to make the participants feel more at ease and willing to share. Instead, we included the study site and population group in the findings to provide the relevant context to the findings. Our findings can be triangulated with the quantitative segment of our mixed-method study [16]. Country-specific teams led by site researchers had maintained rigor for data collection, and thematic analysis of quotes made analysis more systematic and contextual. Relatability to CS was facilitated by its introduction at the start with a video and infographics in local languages.

## Conclusions

In-depth analysis suggested that people are ready to participate with stakeholders such as policymakers and researchers, considering their social duty driven by attitude and culture. Based on their experiences during COVID-19, participation in CS activities viewed as an opportunity to learn new knowledge and scientific skills while working with researchers with some individual level discomforts. Its sustainability can be achieved by establishing organizational system with direct incentives to citizens/participants. Based on current experience, it is recommended that inclusion of optimal and diverse set of participants representing service providers and affected sections of society, especially marginalized and indigenous groups, gives valuable insights. Current methodology can be applied to explore the application of CS in other disease outbreaks/epidemics.

## Supporting information

**S1.Text.  FGD guide.**
(PDF)

**S1 Fig.  Citizen science infographics.**
(PDF)

## Acknowledgments

Authors would like to thank citizens who participated in study along with members of community-based organizations.

**Consent for publication:** As participants provided their informed consent to participate in this study.

Informed consent was also obtained from the individual(s) for the publication of potentially identifiable data included in this article.

## Author contributions

**Conceptualization:** Dinesh Kumar, Yi-Roe Tan, Peiling Yap.

**Data curation:** Dinesh Kumar, Ingo Hauter, Felipe C. Canlas, Firli Yogiteten Sunaryoko, Gyanu Raja Maharjan, Md. Mazharul Anowar, Harjyot Khosa, Yi-Roe Tan.

**Formal analysis:** Dinesh Kumar.

**Funding acquisition:** Peiling Yap.

**Investigation:** Yi-Roe Tan, Peiling Yap.

**Methodology:** Dinesh Kumar, Yi-Roe Tan, Peiling Yap.

**Project administration:** Harjyot Khosa, Yi-Roe Tan.

**Writing – original draft:** Dinesh Kumar.

**Writing – review & editing:** Dinesh Kumar, Ingo Hauter, Felipe C. Canlas, Gyanu Raja Maharjan, Md. Mazharul Anowar, Harjyot Khosa, Yi-Roe Tan, Peiling Yap.

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
