## [Decision Letter · Decision Letter 0]

15 Oct 2024

PGPH-D-24-01917

Qualitative assessment of the citizen science approach to foster public partnerships for pandemic preparedness and response in South and Southeast Asian countries

Dear Dr. Dinesh Kumar,

Thank you for submitting your manuscript to PLOS Global Public Health. After careful consideration, we feel that it has merit but does not fully meet PLOS Global Public Health’s publication criteria as it currently stands. Therefore, we invite you to submit a revised version of the manuscript that addresses the points raised during the review process.

We look forward to receiving your revised manuscript.

Kind regards,

Man Thi Hue Vo, MD, PhD

Academic Editor

Journal Requirements:

 1. Please include a complete copy of PLOS’ questionnaire on inclusivity in global research in your revised manuscript. Our policy for research in this area aims to improve transparency in the reporting of research performed outside of researchers’ own country or community. The policy applies to researchers who have travelled to a different country to conduct research, research with Indigenous populations or their lands, and research on cultural artefacts. The questionnaire can also be requested at the journal’s discretion for any other submissions, even if these conditions are not met.  Please find more information on the policy and a link to download a blank copy of the questionnaire here: https://journals.plos.org/globalpublichealth/s/best-practices-in-research-reporting. Please upload a completed version of your questionnaire as Supporting Information when you resubmit your manuscript. 2. Please amend your detailed Financial Disclosure statement. This is published with the article. It must therefore be completed in full sentences and contain the exact wording you wish to be published. **Please only choose the relevant sentences from below** 1. Please clarify all sources of funding (financial or material support) for your study. List the grants (with grant number) or organizations (with url) that supported your study, including funding received from your institution. 2. State the initials, alongside each funding source, of each author to receive each grant.3. State what role the funders took in the study. If the funders had no role in your study, please state: “The funders had no role in study design, data collection and analysis, decision to publish, or preparation of the manuscript.”4. If any authors received a salary from any of your funders, please state which authors and which funders. 3. In the online submission form, you indicated that "Data can be provided upon request.".  All PLOS journals now require all data underlying the findings described in their manuscript to be freely available to other researchers, either 1. In a public repository, 2. Within the manuscript itself, or 3. Uploaded as supplementary information. This policy applies to all data except where public deposition would breach compliance with the protocol approved by your research ethics board. If your data cannot be made publicly available for ethical or legal reasons (e.g., public availability would compromise patient privacy), please explain your reasons by return email and your exemption request will be escalated to the editor for approval. Your exemption request will be handled independently and will not hold up the peer review process, but will need to be resolved should your manuscript be accepted for publication. One of the Editorial team will then be in touch if there are any issues. 4. We have noticed that you have uploaded Supporting Information files, but you have not included a list of legends. Please add a full list of legends for your Supporting Information files after the references list.  5. Supplementary 2_Citizen science infographics.pdf: Please confirm whether you drew the images / clip-art within the figure panels by hand. If you did not draw the images, please provide (a) a link to the source of the images or icons and their license / terms of use; or (b) written permission from the copyright holder to publish the images or icons under our CC-BY 4.0 license. Alternatively, you may replace the images with open source alternatives. See these open source resources you may use to replace images / clip-art:- https://commons.wikimedia.org-
https://openclipart.org/ 6. Supplementary 1_FGD guide.docx and Supplementary 2_Citizen science infographics.pdf contains branding/a logo. We are not permitted to publish this under our CC-BY 4.0 license, even with permission. We ask that you please remove or replace it.

Additional Editor Comments (if provided):

Major revision requested

Reviewers' comments:

Reviewer's Responses to Questions

**Comments to the Author**

1. Does this manuscript meet PLOS Global Public Health’s publication criteria ? Is the manuscript technically sound, and do the data support the conclusions? The manuscript must describe methodologically and ethically rigorous research with conclusions that are appropriately drawn based on the data presented.

Reviewer #1: Partly

Reviewer #2: Yes

Reviewer #3: Yes

2. Has the statistical analysis been performed appropriately and rigorously?

Reviewer #1: Yes

Reviewer #2: N/A

Reviewer #3: Yes

3. Have the authors made all data underlying the findings in their manuscript fully available (please refer to the Data Availability Statement at the start of the manuscript PDF file)?

Reviewer #1: Yes

Reviewer #2: Yes

Reviewer #3: Yes

4. Is the manuscript presented in an intelligible fashion and written in standard English?

Reviewer #1: Yes

Reviewer #2: Yes

Reviewer #3: Yes

5. Review Comments to the Author

Reviewer #1: Review Report

The manuscript has presented important problem. However, the scope hungs between the community involvement and Research dessimination. The abstract is not understandable and lacks consistency. The background is not focused and lacks strength. The methods section is incomplete for the types of the interview, how the interview was conducted who and how many of the data collectors had participated in the study, the data collection duration, place where the interview was conducted, the numbers of the interview per the interview types. In addition,the soft ware used for the analysis is missed. Above all, trustworthiness of the data is not ensured. The result is not systematically presented.The discussion fails to entail all its contents and strength.

Regards,

Reviewer #2: Introduction

Start by framing the inadequacies of traditional approaches to pandemic management, particularly in terms of community trust and engagement. Then, CS will be introduced as a possible solution to these challenges.

Repetitive Sentences and Ideas: Some ideas are repeated, particularly regarding the importance of community engagement and the potential of CS. Example: The sentence, “CS has piqued the interest of policymakers, program managers, and researchers...,” reiterates points already made earlier about the benefits of CS. Instead, consider combining similar points to streamline the argument.

Provide specific examples of how CS has been used successfully in similar contexts or how it could work in South and Southeast Asia to tackle pandemic-related challenges.

While you mention the need to identify barriers to participation in CS, this section could be expanded. For example, what specific cultural, social, or economic barriers might exist in South and Southeast Asian countries that could impede the successful implementation of CS?

Highlight any gaps in existing research that this study aims to fill. This will underscore the novelty and significance of the research.

Method section

While the methodology is well-explained, it would benefit from additional justification for key decisions. For example, why were 4 focus groups chosen per country? Why was the decision made to exclude participants under 18 years or those with severe mental health conditions?

Some details about the data collection process could be expanded. For instance, how were the focus group discussions conducted? Were they virtual or in person? What measures were taken to ensure participants felt comfortable sharing their thoughts? Additionally, details on how the semi-structured guide based on the Precaution Adoption Process Model and Theory of Planned Behavior shaped the discussion would help the reader understand how these theories guided the research.

The section mentions "thematic saturation" but does not specify how saturation was assessed. This is crucial for qualitative research, as it justifies the number of FGDs conducted.

Reviewer #3: There are some major discrepancies between the larger paper based on this mix-method research carried out by the same authors and published elsewhere (cited in reference 10) that includes five additional countries from Africa and this paper that includes only 5 countries from Asia and claims to be a qualitative study based on the 20 FGDs involving 130 individuals.

Some of these include the following:

1. Definition of CS - the authors now seem to specifically include an additional dimension i.e. "to address public health challenges". They do not describe why the definition for the same study has changed between their publication in BMJ and the current submission.

2. Objectives of the study - This too has changed. how can the same study have different objectives between two separate publications; if the authors on the other hand note that the objectives of the present analysis of the larger study reported already through the BMJ report, then one can understand. But you cannot suddenly have new objectives with wider scope in the same study.

3. Advantages and Disadvantages - Table 4 of the earlier publication (in BMJ) and Table 2 of the present submission in this respect seem to be coming up with different conclusions. To selectively draw only some responses out of 130 individuals and include in a table, appears to be somewhat far-fetched. Further, whereas the earlier publication classifies advantages and disadvantages by digital and analogue communications (and still reports verbatim comments of some respondents), the current report, although meant to be "Qualitative" according to the authors seems to be narrower than the earlier report - without any explanation.

4. Discussion / Conclusions - there seem to be some discrepancies between the discussion and conclusions of the earlier report and the present submission. The authors do not sufficiently explain these differences. Just to cite an example, the earlier report notes: "In Bangladesh, Indonesia, the Philippines, Cameroon and Kenya, majority were unaware of outbreak-related CS"; In the current report - the authors note "Across all countries, while referring to CS, the participants collectively comprehended the term "research". Participants understood that CS is possibly related to capacity building to learn and empower themselves in the field of research"; further they have also noted " It was also viewed as a social responsibility, especially in India. Distinctively, it was referred to as an awareness mechanism to improve knowledge about pandemic response in Bangladesh". There are similar distinctive differences and perceptions of contradictions between the earlier published report and the current manuscript. In my view, the authors have not done a thorough analysis of what they reported in the earlier publication based on the same work and what now they want to publish as a subset and based on FGDs only. At the minimum, they should attempt to quote what they wrote earlier and how and why it is different now.

6. PLOS authors have the option to publish the peer review history of their article (what does this mean? ). If published, this will include your full peer review and any attached files.

**Do you want your identity to be public for this peer review?** For information about this choice, including consent withdrawal, please see our Privacy Policy .

Reviewer #1: No

Reviewer #2: **Yes: ** KHADIJAT ADELEYE

Reviewer #3: **Yes: ** Lakshmi Narasimhan Balaji

---

## [Decision Letter · Decision Letter 1]

7 Jan 2025

PGPH-D-24-01917R1

Qualitative assessment of the citizen science approach to foster public partnerships for pandemic preparedness and response in South and Southeast Asian countries

Dear Dr. Dinesh Kumar,

Thank you for submitting your manuscript to PLOS Global Public Health. After careful consideration, we feel that it has merit but does not fully meet PLOS Global Public Health’s publication criteria as it currently stands. Therefore, we invite you to submit a revised version of the manuscript that addresses the points raised during the review process.

We look forward to receiving your revised manuscript.

Kind regards,

Man Thi Hue Vo, MD, PhD

Academic Editor

Journal Requirements:

Additional Editor Comments (if provided):

Thank you for your recent revisions. While we appreciate the effort to make comprehensive revision, Reviewer 3 has indicated that some critical concerns remain inadequately addressed. To ensure that your manuscript aligns with the expectations of the review process, we request a more detailed explanation and elaboration to address the following specific comments:

1. Reviewer 3 noted discrepancies between Table 4 in your BMJ publication and Table 2 in this manuscript, including differences in conclusions and selective use of responses. Please explain the methodology used for Table 2, clarify why the scope appears narrower, and justify more in details how this analysis builds on or diverges from the earlier report.

2. Discussion/Conclusions discrepancies: Reviewer 3 highlighted differences in how CS was perceived across countries in your BMJ paper and the current submission. Please quote the earlier findings and provide a thorough explanation for these differences with the current findings, emphasizing how this work adds value.

3. From the comments above, Reviewer 3 remains unconvinced that the revision in Lines 372–383 sufficiently shows how this manuscript provides additional detail beyond the BMJ paper. Please expand this discussion with specific and clear examples to strengthen your argument.

4. Please correct typographical errors (e.g., "county" instead of "country" on line 240) and ensure the rest of the manuscript is error-free.

Reviewers' comments:

Reviewer's Responses to Questions

**Comments to the Author**

1. If the authors have adequately addressed your comments raised in a previous round of review and you feel that this manuscript is now acceptable for publication, you may indicate that here to bypass the “Comments to the Author” section, enter your conflict of interest statement in the “Confidential to Editor” section, and submit your "Accept" recommendation.

Reviewer #3: All comments have been addressed

Reviewer #4: All comments have been addressed

2. Does this manuscript meet PLOS Global Public Health’s publication criteria ? Is the manuscript technically sound, and do the data support the conclusions? The manuscript must describe methodologically and ethically rigorous research with conclusions that are appropriately drawn based on the data presented.

Reviewer #3: Yes

Reviewer #4: Yes

3. Has the statistical analysis been performed appropriately and rigorously?

Reviewer #3: I don't know

Reviewer #4: Yes

4. Have the authors made all data underlying the findings in their manuscript fully available (please refer to the Data Availability Statement at the start of the manuscript PDF file)?

Reviewer #3: Yes

Reviewer #4: Yes

5. Is the manuscript presented in an intelligible fashion and written in standard English?

Reviewer #3: Yes

Reviewer #4: Yes

6. Review Comments to the Author

Reviewer #3: While authors have tried to explain their logic, i am not fully convinced with their answers or their response that this is more detailed analysis compared to the BMJ paper. Lines 372 to 383, according to them responds to some of the concerns expressed earlier. But I am not fully convinced.

New mistakes (line 240 - county instead of country) have also crept in.

I am as such unable to make a new recommendation as I believe they do not respond to the concerns expressed adequately

Reviewer #4: The revised paper covers and presents an important topic of interest. All issues raised by previous reviewers have been addressed and i have no further issues with the research paper.

7. PLOS authors have the option to publish the peer review history of their article (what does this mean? ). If published, this will include your full peer review and any attached files.

**Do you want your identity to be public for this peer review?** For information about this choice, including consent withdrawal, please see our Privacy Policy .

Reviewer #3: **Yes: ** Lakshmi Narasimhan Balaji

Reviewer #4: No

---

## [Decision Letter · Decision Letter 2]

18 Mar 2025

PGPH-D-24-01917R2

Qualitative assessment of the citizen science approach to foster public partnerships for pandemic preparedness and response in South and Southeast Asian countries

Dear Dr. Kumar,

Thank you for submitting your manuscript to PLOS Global Public Health. After careful consideration, we feel that it has merit but does not fully meet PLOS Global Public Health’s publication criteria as it currently stands. Therefore, we invite you to submit a revised version of the manuscript that addresses the points raised during the review process.

While the paper has addressed the previous comments of the reviewers, the re-evaluation of the revised article signify the need to address lingering issues. Please refer to the comments of the reviewers for further information.

We look forward to receiving your revised manuscript.

Kind regards,

Dr. Maria Carinnes Alejandria

Academic Editor

Journal Requirements:

Reviewers' comments:

Reviewer's Responses to Questions

**Comments to the Author**

1. If the authors have adequately addressed your comments raised in a previous round of review and you feel that this manuscript is now acceptable for publication, you may indicate that here to bypass the “Comments to the Author” section, enter your conflict of interest statement in the “Confidential to Editor” section, and submit your "Accept" recommendation.

Reviewer #3: All comments have been addressed

Reviewer #5: (No Response)

2. Does this manuscript meet PLOS Global Public Health’s publication criteria ? Is the manuscript technically sound, and do the data support the conclusions? The manuscript must describe methodologically and ethically rigorous research with conclusions that are appropriately drawn based on the data presented.

Reviewer #3: Yes

Reviewer #5: Partly

3. Has the statistical analysis been performed appropriately and rigorously?

Reviewer #3: I don't know

Reviewer #5: N/A

4. Have the authors made all data underlying the findings in their manuscript fully available (please refer to the Data Availability Statement at the start of the manuscript PDF file)?

Reviewer #3: Yes

Reviewer #5: Yes

5. Is the manuscript presented in an intelligible fashion and written in standard English?

Reviewer #3: Yes

Reviewer #5: Yes

6. Review Comments to the Author

Reviewer #3: I am not in a position to make a recommendation one way or another. The authors have attempted to answer concerns partly. But each time, they have not been able to convince me on how this paper is an enhancement of the previous journal paper of their own. As such there appears to be purely some verbiage (semantics) justifying this paper. Does not seem to have any major new information on top of the previous paper. As such leave it for the Editor make an informed judgment of his/her own. [the system will not allow me to say "NO RECOMMENDATION". So even to communicate this to the Editorial team, i have to put in something. I am therefore using the option "MINOR REVISION", although actually it is "NO RECOMMENDATION"

Reviewer #5: 1. Summary of the research

This paper details the findings of focus group discussions in five countries with citizens of each to assess their perceptions of the usefulness, benefits and challenges of using CS for pandemic preparedness and response. While not entirely novel, the findings are interesting and can add value to the current literature on CS and community health. I also enjoyed reading the respondents’ quotations, which provide valuable insights into the topic.

The main argument of this study is that CS might be a viable approach for potential disease prevention efforts by learning from respondents’ experience of the Covid-19 pandemic, and how they believe CS can contribute to future public health crises. This is mentioned in lines 315-318 but is not clearly signposted from the beginning and throughout the paper. The abstract, for instance, makes no specific mention of this.

2. Examples and evidence

Major issues

a) Lack of context on citizen science

This study falls short of providing adequate context for the application of citizen science in pandemic preparedness and response, and in terms of addressing societal issues at the community level. For instance, citizen science has been used with some success to address community health issues in relation to water and air pollution, in environmental monitoring, in food security issues and in disease prevention. Reference to these studies could add context and underscore the significance and benefit of using citizen science for pandemic preparedness.

In the tables, participants refer to CS as being able to overcome geographic and other limitations to data access (Table 1, Participation in Research, Data generation and analysis) and enabling bottom-up research (Table 2, People-centric, Realistic approach). These are among the benefits of CS, which are not adequately discussed in the paper.

b) Lack of justification for the selection of countries

Five countries are included in this paper. However, no explanation is given for why these five countries were chosen. There is a mention of this study being part of a larger study, findings of which has been published in a separate paper (see line 109). In that paper, it was mentioned that these countries were chosen at least partly to fill the gap in the literature, which has so far focused on high income countries. It would be beneficial to include the same information in this paper.

c) Lean into the demographic of the participants

There is an opportunity here to highlight that the demographic of the participants in this study (the youths and marginalized and indigenous populations) fills the gap in many CS studies, which have shown that these groups are generally excluded from CS as citizen participants as well as the subjects of CS projects. Please refer to published papers that have examined the demographics of CS volunteers.

Minor issues

a) Include quotations in the findings and discussions rather than highlighting them only in the tables, to make the texts more engaging and interesting.

b) There are some sentences that are truncated and incomplete. See e.g lines 87-90, 91-92 and 94-97.

c) Unclear focus on youth

In line 114-115 it is mentioned that the study involved youths (participants were purposively sampled to include youths…”) as well as other groups. However in line 121-123, youths were highlighted as if they were the sole focus of the study (“As study focused on youths…”). This inconsistency can be confusing and misleading.

d) General population

Unclear what the general population who make up the study sample refer to. Please clarify.

e) The study refers to the infographic and videos used to explain to participants what CS is. These are linked in the document to external resources. For the benefit of text readers, it would be useful to include a brief description of the content of these materials within the text.

f) Title of the paper could perhaps be clearer about the focus of the study. “Qualitative assessment of the CS approach…” is quite vague. “Community perceptions of the CS approach in pandemic preparedness” provides the reader with immediate and clear information about the study.

7. PLOS authors have the option to publish the peer review history of their article (what does this mean? ). If published, this will include your full peer review and any attached files.

**Do you want your identity to be public for this peer review?** For information about this choice, including consent withdrawal, please see our Privacy Policy .

Reviewer #3: **Yes: ** Lakshmi Narasimhan Balaji

Reviewer #5: No

---

## [Decision Letter · Decision Letter 3]

6 May 2025

PGPH-D-24-01917R3

Community perceptions of citizen science approach in pandemic preparedness and response in South and Southeast Asian countries

Dear Dr. Dinesh Kumar,

Thank you for submitting your manuscript to PLOS Global Public Health. After careful consideration, we feel that it has merit but does not fully meet PLOS Global Public Health’s publication criteria as it currently stands. Therefore, we invite you to submit a revised version of the manuscript that addresses the points raised during the review process.

Overall

The authors are requested to prepare the response file by showing the revised changes directly in the response file, instead of only providing line numbers. This will facilitate the review of the changes made in the manuscript. In case of major changes, line numbers may be mentioned only.It would be helpful to avoid abbreviations throughout the document, as they interrupt the reading flow.

Introduction

Lines 85 to 87: Please expand the term 'diverse geographical settings.' It would be helpful to mention the names of the geographical settings referenced here.Line 88 to 90: lease expand the sentence further. From the current description, it is unclear what significant variations you are referring to and how they are connected to your paper."

Methods

The Methods section should include separate paragraphs for each theme, for example, study site and design, participant selection, data collection, etc. This will improve the clarity of the Methods section, which is currently somewhat difficult to follow. It is also important to ensure transferability of this research.  The participant selection criteria need to be clarified further. While the type of group is defined, it is not clear how and from where the participants were selected. Although the study country is specified, it is important to explain how the researchers chose the specific study location and how participant selection was carried out. In qualitative research, providing a clear description of participant selection is essential to ensure rigor and support transferability.A survey is mentioned in the Methods section; providing a bit more detail about this survey would help readers better understand the context. It would also clarify the participant selection process.It is stated that the staff who collected data had good rapport with the community, but it is not clear how this rapport was developed or what the process of becoming familiar with participants involved. Additionally, since participants were interviewed in a hall or health center, it should be clarified how a ‘natural setting’ was maintained during the interviews.The information on data saturation should be included in the analysis section rather than in the description of the tools.The Methods section mentions the use of PAPM and TPB; however, a more detailed explanation linking these frameworks to the objective of the manuscript is important. For example, in lines 167 and 168, 'a series of stages' is mentioned, but it is unclear how this was connected to the FGD guides, the analysis, or how it is presented in the results. Elaborating on this would help demonstrate better relevance.Lines 170 to 184 mention 'awareness, relatability, acceptability, feasibility, and sustainability,' but the descriptions in the supplementary file differ. It would be helpful to provide a brief description of the themes and subthemes in the Methods section rather than referring only to the supplementary tables.Lines 178 to 179: The sentence 'Venues to conduct FGD…' should be moved to the section where venue-related information is provided (from line 160 onwards).Lines 180 to 184: It would be helpful to define or specify the 'various stages' of citizen engagement.The analysis procedures followed should be clearly described, with appropriate references, rather than simply mentioning 'content analysis’.It is also important to specify the type of consent obtained—whether written or verbal.

Results

A table presenting the demographic profile of all participants is required; otherwise, it is difficult to connect the findings to the participants’ context. The demographic information may include age, education, occupation, or any other background details relevant to understanding the context of their responses.Lines 221 to 222: The sentence 'Domains of FGD are summarized by the theme…' should be moved to the analysis section of the Methods, as it reflects methodological details rather than results.Lines 232 to 235: The terminology 'awareness mechanism' is unclear; it would be helpful if the authors could revise or clarify its meaning.In several places in the Results section, terms such as 'majority' and 'most' are used. It is important to define these terms—either in the Methods section or in the Results—by indicating the number of responses out of the total participants in parentheses.

Discussion

The first sentence of the Discussion should present the main findings connected to the main objective of the research (mentioned in last sentence of the introduction).Lines 333 to 338: This sentence will be better justified if the authors provide a clear description of the rationale for participant selection in the Methods section and connect the results more explicitly to these different categories.Lines 340 to 350: The sentence 'it is mentioned that… in analogue form' seems more like a rationale and should be moved to the Introduction section.Lines 360 to 378: This description appears to be a repetition of the results. It would be helpful if the authors could link the findings with existing literature and discuss them further.Lines 439 to 446: To describe this adequately, a detailed and clear explanation of the participant selection methods is crucial. Without this, the rigor of the study will be difficult to assess.Lines 466 to 469: The recommendation regarding methodological strengths discussed throughout the Discussion section is missing. The authors may consider adding it alongside the current recommendations.

Tables

The term 'Sir' is mentioned in several places, indicating a hierarchy between the participants and the researcher. This suggests that rapport building may not have been optimal, and the natural setting was interrupted, which could have led to biased responses. It would be helpful to clarify this in the Methods section and address it as a limitation.

We look forward to receiving your revised manuscript.

Kind regards,

Rebeca Sultana

Academic Editor

Journal Requirements:

Additional Editor Comments (if provided):

Reviewers' comments:

Reviewer's Responses to Questions

**Comments to the Author**

1. If the authors have adequately addressed your comments raised in a previous round of review and you feel that this manuscript is now acceptable for publication, you may indicate that here to bypass the “Comments to the Author” section, enter your conflict of interest statement in the “Confidential to Editor” section, and submit your "Accept" recommendation.

Reviewer #3: All comments have been addressed

Reviewer #5: All comments have been addressed

2. Does this manuscript meet PLOS Global Public Health’s publication criteria ? Is the manuscript technically sound, and do the data support the conclusions? The manuscript must describe methodologically and ethically rigorous research with conclusions that are appropriately drawn based on the data presented.

Reviewer #3: Yes

Reviewer #5: Yes

3. Has the statistical analysis been performed appropriately and rigorously?

Reviewer #3: I don't know

Reviewer #5: N/A

4. Have the authors made all data underlying the findings in their manuscript fully available (please refer to the Data Availability Statement at the start of the manuscript PDF file)?

Reviewer #3: No

Reviewer #5: Yes

5. Is the manuscript presented in an intelligible fashion and written in standard English?

Reviewer #3: Yes

Reviewer #5: Yes

6. Review Comments to the Author

Reviewer #3: I continue to have difficulty with the paper in most parts being a part reporting of an earlier paper cited in reference 17 by the authors. As such it does not completely note why this subset is different or necessary out of the total of 9 countries already reported in the BMJ paper.

Reviewer #5: (No Response)

7. PLOS authors have the option to publish the peer review history of their article (what does this mean? ). If published, this will include your full peer review and any attached files.

**Do you want your identity to be public for this peer review?** For information about this choice, including consent withdrawal, please see our Privacy Policy .

Reviewer #3: **Yes: ** Lakshmi Narasimhan Balaji

Reviewer #5: No

---

## [Editor Report · Decision Letter 4]

4 Jul 2025

PGPH-D-24-01917R4

Community perceptions of citizen science approach in pandemic preparedness and response in South and Southeast Asian countries

Dear Dr. Kumar,

Thank you for submitting your manuscript to PLOS Global Public Health. After careful consideration, we feel that it has merit but does not fully meet PLOS Global Public Health’s publication criteria as it currently stands. Therefore, we invite you to submit a revised version of the manuscript that addresses the points raised during the review process.

In the previous decision letter about this manuscript, reviewer 3 commented the following: 

"I continue to have difficulty with the paper in most parts being a part reporting of an earlier paper cited in reference 17 by the authors. As such it does not completely note why this subset is different or necessary out of the total of 9 countries already reported in the BMJ paper."

In your response to reviewers, please directly address the overlap between the present manuscript and your previous study (17. Tan YR, Nguyen MD, Mubaira CA, et al. Building citizen science intelligence for outbreak preparedness and response: A mixed-method study in nine countries to assess knowledge, readiness and feasibility. BMJ Glob Health. 2024;9(3):e014490). Please also ensure that you revise your manuscript where necessary to better clarify the distinction between the two papers, given our policy that: "We strongly discourage the unnecessary division of related work into separate manuscripts, and we will not consider manuscripts that are divided into parts. Each submission to PLOS must be written as an independent unit and should not rely on any work that has not already been accepted for publication."

We look forward to receiving your revised manuscript.

Kind regards,

Sarah Jose, Ph.D.

Staff Editor
---

## [Editor Report · Decision Letter 5]

14 Jul 2025

PGPH-D-24-01917R5

Community perceptions of citizen science approach in pandemic preparedness and response in South and Southeast Asian countries

Dear Dr. Kumar,

Thank you for submitting your manuscript to PLOS Global Public Health. After careful consideration, we feel that it has merit but does not fully meet PLOS Global Public Health’s publication criteria as it currently stands. Therefore, we invite you to submit a revised version of the manuscript that addresses the points raised during the review process.

Request from the Editorial Office: Thank you for your clarification of the differences between your previous study published in the BMJ and the present study. Can you please include a discussion of these differences in the Introduction and perhaps Discussion of the present manuscript? It is important to clarify the context and the knowledge gap filled by your present work. Please feel free to contact us at globalpubhealth@plos.org (please cc myself, sjose@plos.org, for a faster response). Thank you for your attention to this request.

We look forward to receiving your revised manuscript.

Kind regards,

Sarah Jose, Ph.D.

Staff Editor
---

## [Editor Report · Decision Letter 6]

18 Jul 2025

PGPH-D-24-01917R6

Community perceptions of citizen science approach in pandemic preparedness and response in South and Southeast Asian countries

Dear Dr. Kumar,

Thank you for submitting your manuscript to PLOS Global Public Health. After careful consideration, we feel that it has merit but does not fully meet PLOS Global Public Health’s publication criteria as it currently stands. Therefore, we invite you to submit a revised version of the manuscript that addresses the points raised during the review process.

Thank you very much for your revisions. We still feel your distinctions are not sufficiently clarified in the manuscript itself. I kindly request that you address the following points:

1. In the Introduction, from line 95 to 99, please clarify that you are describing your own previous work. This is important for transparency, and to provide the background for why you are now performing this related study to further elucidate some of the knowledge gaps that remain.

2. In your Discussion, your newly added text is a nice explanation, but it is not clear that the preceding sentence is a description of your own previous work. Consider changing "As a part of study" to "In our previous study,[ref 17]". Can you describe the findings of the current study and how they relate to those of the previous one at all? 

Thank you for helping us to ensure the two papers are clearly differentiated in your text.

We look forward to receiving your revised manuscript.

Kind regards,

Sarah Jose, Ph.D.

Staff Editor
---

## [Editor Report · Decision Letter 7]

22 Jul 2025

Community perceptions of citizen science approach in pandemic preparedness and response in South and Southeast Asian countries

PGPH-D-24-01917R7

Dear Professor Kumar,

We are pleased to inform you that your manuscript 'Community perceptions of citizen science approach in pandemic preparedness and response in South and Southeast Asian countries' has been provisionally accepted for publication in PLOS Global Public Health.

Best regards,

Julia Robinson

Staff Editor